# CamGeo: Sparse Camera-Conditioned Image-to-Video Generation with 3D Geometry Priors

**Xuanyi Liu** [1] **Deyi Ji** [2] **Liqun Liu** [2] **Lanyun Zhu** [3] **Xuhang Chen** [4] **Qianxiong Xu** [5] **Peng Shu** [2] **Huan Yu** [2] **Jie Jiang** [2] **Feng Gao** [1] **Siwei Ma** [1]

## Abstract

Sparse camera-conditioned image-to-video generation presents a pivotal challenge: synthesizing geometrically consistent 3D motion from minimal pose cues. Existing methods, which largely rely on dense supervision or naive interpolation, suffer from severe pose drift and motion discontinuities due to the lack of robust 3D priors. In this paper, we introduce **CamGeo**, a novel framework that distills rich 3D geometric knowledge from a pre-trained video-to-3D model (VGGT) directly into the diffusion backbone. To achieve this without incurring inference latency, we propose a training-only distillation strategy. Specifically, CamGeo incorporates: (1) keyframe trajectory distillation that enforces cycle-consistency with sparse input poses, (2) cross-frame consistency distillation with both camera trajectory and depth constraints to generate consistent structure across unsupervised frames, and (3) a three-stage coarse-to-fine curriculum learning, progressively scales geometric complexity, from global structure coherence to fine-grained refinement, achieving stable optimization. Extensive experiments demonstrate that CamGeo achieves consistent improvements under various sparsity ratios.

## 1. Introduction

The pursuit of controllable image-to-video (I2V) generation (Guo et al., 2023; 2024a; Ren et al., 2024; Zhao et al., 2024) has positioned camera-conditioned synthesis as a highly practical and expressive paradigm, enabling directo-

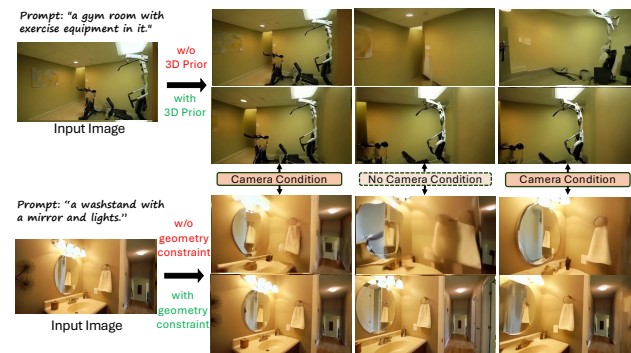

*Figure 1.* Top: Under sparse camera conditioning, interpolating missing poses without geometry priors leads to severe pose drift and structural collapse (e.g., distorted equipment). Bottom: By integrating 3D geometric constraints, our method produces consistently stable and plausible video sequences.

rial control over viewpoint trajectories. However, existing methods (Xu et al., 2024; Zheng et al., 2024; He et al., 2024) rely on dense, per-frame camera poses, which can be difficult to obtain for in-the-wild videos. In practice, classical 3D reconstruction pipelines (Wei et al., 2020; Wu et al., 2011) such as COLMAP (Schönberger et al., 2016; Schönberger & Frahm, 2016) may struggle to produce stable and temporally consistent poses when facing fast motion or complex non-rigid dynamics, making such dense pose supervision hard to ensure. A more user-friendly and broadly applicable approach is sparse camera control, where generation is guided by only a few keyframes. Yet, this direction remains critically underexplored; current methods, even those like CMG (Cheong et al., 2024) that support sparse inference, are still trained on densely annotated datasets like RealEstate10K (Zhou et al., 2018), creating a fundamental disconnect between their training objective and real-world application needs. This raises a pivotal question: *can a model learn to generate high-quality, consistent videos by being trained directly on sparse camera conditions from the outset, mirroring how it will ultimately be used?*

A standard approach to bridge the sparsity gap is to interpolate missing camera poses via mathematical heuristics (e.g., SLERP) and condition the generation on this pseudo-dense

Xuanyi Liu completed this work while interning at Tencent as part of the Tencent Rhino-Bird Research Elite Program, with Deyi Ji as the program leader. [1]School of Computer Science, Peking University [2]Tencent [3]Tongji University [4]University of Cambridge [5]Nanyang Technological University. Correspondence to: Deyi Ji <jideyi16@foxmail.com>, Siwei Ma <swma@pku.edu.cn>.

trajectory (Guo et al., 2024b; Xi et al., 2025). However, as illustrated in Fig.1, this strategy is constrained by two critical issues. First, lacking robust internal 3D priors, the model exhibits significant *pose drift* on frames without explicit conditioning, producing content that defies physical laws (Fig.1, Top). Second, rigid mathematical interpolation fails to capture the non-linear dynamics of real-world camera motion (e.g., handheld shake), resulting in *jerky and discontinuous motion*. Fundamentally, these artifacts arise because the model is tasked with "hallucinating" 3D geometry without feedback on the validity of its hallucinations, leading to a disconnect between the conditioning signal and the generated frames.

In this paper, we propose a novel framework named **Cam-Geo**, which effectively addresses the aforementioned issues and achieves significantly improved performance. Our core idea is to leverage the rich geometric priors from VGGT, a pre-trained model that excels at inferring comprehensive 3D scene attributes from sparse views, to guide the video diffusion model under sparse camera conditions. This approach refines the model's inherent 3D understanding, enabling it to *hallucinate geometrically consistent and smooth camera trajectories* from minimal inputs, thereby producing videos with higher visual quality and physical plausibility. However, directly injecting VGGT's high-dimensional geometry features into the video backbone is unsuitable, as it introduces prohibitive computational overhead during inference.

Therefore, we propose a novel *training-only distillation strategy*. We use VGGT to provide two key supervisory signals during training: (1) Keyframe Trajectory Distillation, ensuring strict adherence to the provided sparse poses, and (2) Cross-frame Consistency Distillation, enforcing smooth and coherent geometry in unconstrained frames. Crucially, this powerful 3D guidance is only required during training and is completely removed at inference, maintaining high efficiency while achieving superior generation quality.

Furthermore, optimizing the diffusion model with these two distillation mechanisms presents a significant challenge: the model concurrently masters macroscopic camera motion and microscopic geometric details. To ensure stable convergence and high-quality output, we propose a coarse-to-fine curriculum learning strategy that aligns with the diffusion model's inherent generative nature. The curriculum consists of three phases: (1) A warm-up stage to let the model learn to establish basic visual coherence and temporal continuity. (2) the coarse-grained phase, we focus on establishing global structural coherence through strong camera pose constraints, allowing the model to first learn stable viewpoint trajectories. (3) the fine-grained phase, the emphasis shifts to geometric refinement, where depth-based supervision is progressively intensified to enforce detailed 3D consistency while relaxing pose constraints. The transition between phases is governed by a smooth sigmoidal scheduling function that adapts loss weights throughout training. This structured progression enables the model to robustly capture holistic motion patterns before addressing fine-grained spatial relationships, leading to improved stability and superior generation quality.

In conclusion, our contributions are threefold: (1) We propose CamGeo, a novel framework that leverages VGGT's 3D geometric priors for sparse camera-conditioned video generation, employing a training-only distillation strategy that eliminates inference overhead. (2) We introduce a coarse-to-fine curriculum learning approach that progressively scales geometric complexity, ensuring stable convergence from global motion to local details. (3) Extensive experiments demonstrate improved performance across various sparse conditioning scenarios, providing an efficient solution for practical video generation applications.

## 2. Related Work

**Camera Conditioned Video Generation.** Early controllable video generation methods focused on learning motion priors. AnimateDiff (Guo et al., 2023) introduced a plug-and-play motion module, while MotionCtrl (Wang et al., 2024b) disentangled camera and object motion for fine-grained control. Recent works have explored camera-conditioned video generation. CameraCtrl (He et al., 2024) and CamCo (Xu et al., 2024) incorporated camera inputs for geometric consistency. CamI2V (Zheng et al., 2024) used epipolar attention for viewpoint consistency, and CPA (Wang et al., 2024a) employed pose-aware attention in Diffusion Transformers. VD3D (Bahmani et al., 2024) integrated camera embeddings via Plücker coordinates, while Wan2 (Wan et al., 2025) encoded camera parameters for motion alignment. However, these methods require dense per-frame camera supervision from high-frame-rate capture and expensive reconstruction. In contrast, we address the more challenging sparse camera-conditioned setting with only a few keyframes. To bridge this information gap, we leverage 3D priors for smooth, geometrically coherent motion generation.

**Sparse Conditioned Video Generation.** Recent research has explored sparse conditioning as a means to reduce computation and improve controllability in video generation(Xi et al., 2025). SparseCtrl (Guo et al., 2024b) demonstrated that temporally sparse structure cues such as sketches, depth, or RGB keyframes can effectively guide text-to-video models without modifying the base generator. YODA (Davtyan & Favaro, 2024) introduced an unsupervised framework that autoregressively generates videos from a single frame and sparse motion cues, implicitly learning object interactions and dynamics from randomized conditioning. Similarly, RIVER (Davtyan et al., 2023) leveraged latent flow matching conditioned on a random subset of past frames to achieve

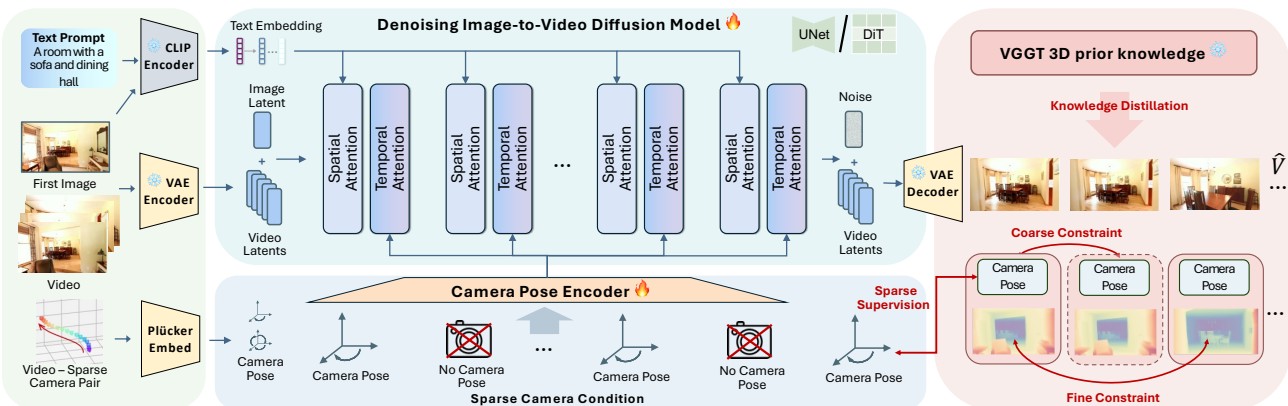

*Figure 2.* Overview of the CamGeo Framework. Under sparse camera conditioning, our framework distills 3D geometric priors from a pre-trained, frozen VGGT into a denoising diffusion model (supporting both UNet and DiT architectures). During training, the decoded clean video $\hat{V}$ is processed by VGGT to extract camera trajectories and geometric constraints, which provide multi-stage supervision through a progressive curriculum. For visual clarity, the DiT patchify block is omitted.

efficient video prediction, while Sparse VideoGen (Xi et al., 2025) revealed intrinsic sparsity in the attention patterns of diffusion transformers, enabling acceleration of large-scale video generation. MagicMotion (Li et al., 2025) extended sparsity into the trajectory-control domain, supporting motion guidance through sparse spatial inputs such as bounding boxes or keypoints, yet primarily focused on object-level motion.

**3D Geometry Priors with VGGT.** The emergence of feed-forward 3D foundation models has revolutionized geometric understanding in computer vision. Among these, VGGT (Wang et al., 2025b) stands out as a pioneering framework that jointly infers comprehensive 3D scene attributes, including camera parameters, depth maps, and 3D point tracks, from unconstrained image sets in a single forward pass. Recent works have successfully leveraged VGGT's powerful priors across diverse domains (Liu et al., 2025; Wang et al., 2025a; Deng et al., 2025; Shen et al., 2025; Zhang et al., 2025; Vuong et al., 2025; Zhou et al., 2025; Zhuo et al., 2025; Qian et al., 2025; Liu et al., 2026). VGGT-X (Liu et al., 2025) demonstrated its effectiveness in dense novel view synthesis, achieving COLMAP-free state-of-the-art performance. In robotic manipulation, VGGT-DP (Ge et al., 2025) integrated geometric priors with proprioceptive feedback for precise visuomotor control. Despite these advancements, the potential of VGGT priors remains unexplored in camera-conditioned image-to-video generation.

## 3. Method

### 3.1. Problem Formulation

We address the problem of sparse camera-conditioned image-to-video (I2V) generation. Given a reference image and a text prompt, the objective is to synthesize a high-

fidelity video sequence $V = \{I_f\}_{f=1}^{F}$ that adheres to a sparse set of camera poses defined on a subset of frames $\mathcal{S} \subset \{1, \ldots, F\}$, where $|\mathcal{S}| \ll F$, and sparsity ratio is defined as $r = \frac{|\mathcal{S}|}{F} \in (0, 1)$. The camera condition is denoted as $c_{\text{cam}} = \{\mathbf{E}_s, \mathbf{K}_s\}_{s \in \mathcal{S}}$, comprising the extrinsic matrix $\mathbf{E}_s \in \mathbb{R}^{3 \times 4}$ and intrinsic matrix $\mathbf{K}_s \in \mathbb{R}^{3 \times 3}$ for the $s$-th frame.

Our method builds on a pre-trained text-guided latent diffusion I2V model. Given latent video clips $z_0^{1:F}$, we corrupt them with Gaussian noise over diffusion steps $t \in \{1, \ldots, \mathcal{T}\}$ to obtain $z_t^{1:F}$. We denote the full conditioning signal as

$$c = \{c_{\text{text}}, c_{\text{img}}, c_{\text{cam}}\}, \quad (1)$$

where $c_{\text{text}}$ is the prompt, $c_{\text{img}}$ is the reference image condition, and $c_{\text{cam}}$ provides pose constraints only at frames in $\mathcal{S}$. Following prior work (He et al., 2024), we encode camera poses using Plücker embeddings (Sitzmann et al., 2021). We train a denoiser $\epsilon_\theta$ to predict the injected noise $\epsilon$:

$$\mathcal{L}_{\text{diff}} = \mathbb{E}_{z_0^{1:F}, c, \epsilon, t}\left[\left\|\epsilon - \epsilon_\theta(z_t^{1:F}, c, t)\right\|_2^2\right]. \quad (2)$$

The key challenge departs from standard conditional generation: most viewpoints are unconstrained. The model must therefore infer a geometrically consistent and physically plausible camera motion between sparsely specified, arbitrarily located keyframes, while maintaining visual fidelity, temporal coherence, and 3D consistency.

### 3.2. 3D Geometry Priors as Knowledge

A central difficulty in sparse camera conditioning ($|\mathcal{S}| \ll F$) is that the model must synthesize geometrically consistent novel views for the vast majority of frames without explicit

pose supervision. While recent advances in 3D understanding (Zhang et al., 2025; Yang et al., 2025) demonstrate that VGGT serves as a powerful feature extractor for geometry-aware representation learning, integrating it directly into the generation pipeline would require execution at inference time, introducing substantial latency and memory overhead.

To equip the model with robust 3D reasoning without altering its efficient inference architecture, we propose a training-only distillation strategy (Fig. 2). We leverage a pre-trained VGGT model as a frozen teacher to provide multi-level geometric supervision solely during training, which is entirely removed at test time. To enable geometric consistency losses within the diffusion training loop, we utilize the model's predicted clean video $\hat{V}$ at each timestep. Specifically, we estimate the clean latent $\hat{z}_0$ from the noisy input $z_t$ and predicted noise $\epsilon_\theta(z_t)$, and decode it to pixel space via the scheduling parameter $\bar{\alpha}_t$ (Ho et al., 2020) and pre-trained decoder $\mathcal{D}$:

$$\hat{z}_0 = \frac{z_t - \sqrt{1 - \bar{\alpha}_t}\epsilon_\theta(z_t, t)}{\sqrt{\bar{\alpha}_t}}, \quad \hat{V} = \mathcal{D}(\hat{z}_0). \quad (3)$$

Given the predicted video $\hat{V} = \{\hat{I}_f\}_{f=1}^F$, the frozen VGGT teacher estimates dense camera trajectories $\hat{\mathbf{C}} = \{\hat{\mathbf{R}}_f, \hat{\mathbf{T}}_f, \hat{\mathbf{K}}_f\}_{f=1}^F$ and per-frame depth maps $\hat{\mathbf{D}} = \{\hat{D}_f\}_{f=1}^F$. We distill these geometric cues into the diffusion model through two complementary objectives targeting different temporal domains: (i) **Keyframe Trajectory Distillation**, which acts on the *sparse keyframes* ($s \in \mathcal{S}$) to anchor the generated structure to the condition viewpoints; and (ii) **Cross-frame Consistency Distillation**, which operates across the *temporal intervals* to propagate geometric consistency from the anchors to the unsupervised intermediate frames.

### 3.2.1. KEYFRAME TRAJECTORY DISTILLATION

During training, we feed the predicted video $\hat{V} = \{\hat{I}_f\}_{f=1}^F$ to the frozen VGGT teacher to obtain a dense camera estimate $\hat{\mathbf{C}} = \{\hat{\mathbf{R}}_f, \hat{\mathbf{T}}_f, \hat{\mathbf{K}}_f\}_{f=1}^F$. We use the sparse conditioning frames $\mathcal{S}$ as *geometric anchors* and enforce a cycle-consistency constraint between the VGGT-estimated parameters and the provided ground-truth camera poses. Recognizing the need for a robust and efficient representation for 3D rotations, we employ unit quaternions (Kendall et al., 2015) to measure the discrepancy in camera orientation:

$$\mathcal{L}_{\text{traj}} = \sum_{s \in \mathcal{S}} \left( \|\phi(\hat{\mathbf{R}}_s) - \phi(\mathbf{R}_s)\|_1 + \|\hat{\mathbf{T}}_s - \mathbf{T}_s\|_1 + \|\hat{\mathbf{K}}_s - \mathbf{K}_s\|_1 \right),$$

$$(4)$$

where $\phi(\cdot)$ maps a rotation matrix to a unit quaternion, and $(\mathbf{R}_s, \mathbf{T}_s, \mathbf{K}_s)$ denotes condition camera poses at frame $s$.

This formulation offers three advantages: 1) The $\mathcal{L}_1$ norm provides a robust optimization landscape, reducing the ad-

verse influence of inherent estimation errors from the teacher model and enhancing training stability; 2) The use of quaternions enables numerically stable rotation comparison while avoiding singularities associated with rotation matrices; 3) It establishes a direct, self-supervised cycle that anchors the generated video's geometry to the user's sparse inputs. This ensures that for any frame with a known camera pose, the model's output is rigorously aligned, effectively preventing catastrophic pose drift and ensuring faithfulness to the control signal.

### 3.2.2. CROSS-FRAME CONSISTENCY DISTILLATION

Under sparse camera conditioning, frames without explicit supervision ($f \notin \mathcal{S}$) are prone to severe geometric drift. A key limitation of local-only smoothing is that intermediate frames (e.g., frame 2 in a sequence where only 0 and 4 are anchored) remain isolated from reliable anchors, leading to error accumulation and implausible trajectories.

Interestingly, our empirical analysis shows that the SVD-Full baseline, even without explicit geometry, outperforms interpolation-based methods, whose results are shown in Table 1). We attribute this to the fact that rigid mathematical interpolation (e.g., SLERP) often violates physical motion priors, whereas generative models leverage learned internal priors to perform "semantic interpolation" that better captures natural transitions.

Building on this insight, we propose a **Geometry Smoothness Distillation** strategy to bridge the gap between rigid geometry and generative priors. Specifically, we employ the VGGT teacher to estimate continuous camera parameters $\hat{\mathbf{C}}$ and depth maps $\hat{\mathbf{D}}$ from the generated sequence $\hat{\mathbf{V}}$. To enforce geometric coherence across moving viewpoints, we avoid naive temporal depth subtraction and instead employ a geometry-aware warping mechanism. We project the geometry of frame $f$ into the reference frame $f + k$ in the keyframe interval ($s_i \leq f, f + k \leq s_{i+1}$). To account for the scale and shift ambiguity inherent in monocular estimators, we apply a scale-invariant transformation (Ranftl et al., 2020) to the depth maps before computing loss:

$$\mathcal{L}_{\text{geo}} = \sum_{f,k} \lambda^{(k)} w_{f,f+k} \Big( \underbrace{\|\hat{D}_{f+k} - \mathcal{W}(\hat{D}_f, \Delta\hat{\mathbf{E}}_{f,f+k}, \hat{\mathbf{K}})\|_1}_{\text{Geometric Depth Consistency}}$$

$$+ \underbrace{\|\Delta(\hat{\mathbf{C}}_{f+k}, \hat{\mathbf{C}}_f)\|_1}_{\text{Trajectory Smoothness}} \Big)$$

$$(5)$$

where $\mathcal{W}$ denotes bilinear warping, $\Delta\hat{\mathbf{E}}_{f,f+k}$ the relative pose, estimated $\hat{\mathbf{K}}$ parameter remains consistent in a video, and $\Delta(\cdot)$ the pose distance. The stride selector $\lambda^{(k)} \in \{0, 1\}$ toggles between local ($\lambda^{(1)}$) and long-range ($\lambda^{(k>1)}$) constraints. To balance these contributions, we define a

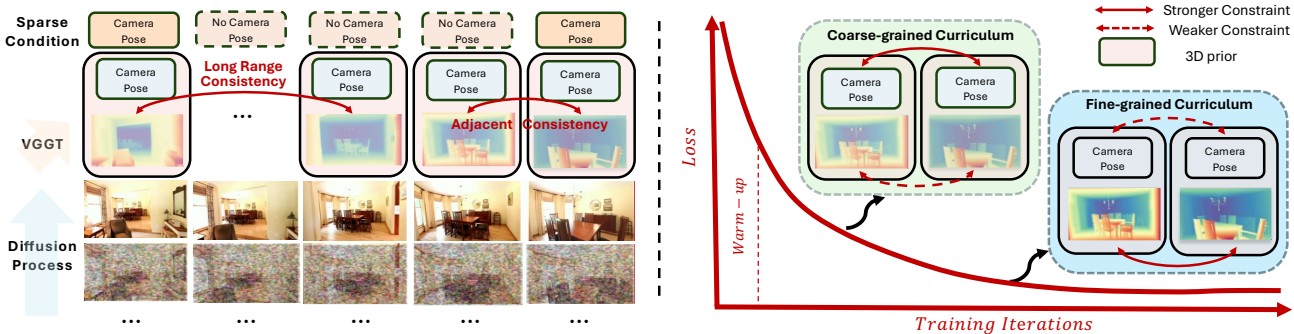

*Figure 3.* (Left) Under sparse camera conditioning, VGGT provides training-only 3D supervision via trajectory and smoothness distillation. (Right) A coarse-to-fine curriculum transitions the training focus from warm-up to strong pose smoothness, and then to detailed depth refinement for stable optimization.

dynamic weight $w_{f,f+k}$:

$$w_{f,f+k} = \exp(\gamma \cdot k) \cdot \exp(-\eta \|\nabla \hat{I}_f\|_1). \quad (6)$$

This weight incorporates: (1) **Long-range Anchoring** ($\exp(\gamma k)$), which prioritizes larger temporal gaps to propagate anchors and prevent trajectory drift; and (2) **Content Adaptivity** ($\exp(-\eta \|\nabla \hat{I}_f\|_1)$), which reduces penalties in high-gradient or occluded regions to mitigate warping artifacts. This multi-scale distillation ensures global 3D coherence.

### 3.3. Coarse-to-Fine Curriculum Conditioning

While VGGT teacher has provided a realiable physical knowledge, optimizing a diffusion model to strictly adhere to sparse geometric constraints presents two fundamental challenges: the reliability of the supervisory signal and the complexity of the optimization landscape.

As detailed in Appendix A, we empirically observe that at the initial stages of training, the generated videos $\hat{V}$ contain severe artifacts and temporal inconsistencies. Feeding these low-quality, out-of-distribution inputs to the frozen VGGT teacher results in unreliable pose and depth estimates, which can propagate noisy gradients and destabilize the diffusion training. Further, applying all constraints simultaneously often leads to training instability, as the model is overwhelmed by conflicting gradients, striving for pixel-level geometric precision while simultaneously learning global viewpoint consistency.

To resolve these problems, we design a coarse-to-fine **three-phase curriculum strategy** that progressively introduces geometric constraints from global to refinement:

**Phase 1: Warm-up (Basic Structure).** In the initial training iterations ($t_{\text{train}} < T_{\text{warm}}$), the priority is to establish basic visual coherence and temporal continuity. During this phase, the VGGT-based distillation losses are deactivated

($\lambda_{\text{traj}} = \lambda_{\text{smth}} = 0$) to prevent noisy gradients from confusing the model. The model is trained solely with the standard diffusion loss $\mathcal{L}_{\text{diff}}$ to learn the data distribution and generate structurally plausible videos.

**Phase 2: Coarse-grained (Global Structural Stabilization).** Once the model produces visually coherent frames ($t_{\text{train}} \geq T_{\text{warm}}$), we activate the trajectory distillation. The model initially focus on the global structure following the camera motion. We apply a strong camera trajectory constraint while keeping the depth-based warping loss low-weighted. This forces the model to learn valid camera kinematics without being overwhelmed by pixel-level geometric errors.

**Phase 3: Fine-grained (Geometric Refinement).** In the third phase, as the trajectory stabilizes, we transition to fine-grained refinement. We gradually introduce the warping-based depth consistency loss to enforce dense 3D surface alignment.

**Curriculum Scheduling.** We formalize this strategy using a global reliability weight $\alpha$ and a refinement factor $\beta$, and we also reformalize Eq.5, which are functions of the training iteration $t_{\text{train}}$:

$$\mathcal{L}_{\text{geo}} = \sum_{f,k} \lambda^{(k)} w_{f,f+k} \big[ \beta \|\hat{D}_{f+k} - \mathcal{W}(\hat{D}_f, \Delta\hat{\mathbf{E}}_{f,f+k}, \hat{\mathbf{K}}_f)\|_1 $$
$$+ (1-\beta)\|\Delta(\hat{\mathbf{C}}_{f+k}, \hat{\mathbf{C}}_f)\|_1 \big]$$
$$\mathcal{L}_{\text{total}} = \mathcal{L}_{\text{diff}} + \alpha \cdot [(1-\beta)\mathcal{L}_{\text{traj}} + \mathcal{L}_{\text{geo}}]$$
$$\text{where } \alpha = \text{Sigmoid}\left(\frac{t_{\text{train}} - T_{\text{warm}}}{\tau_1}\right), \beta = \text{Sigmoid}\left(\frac{t_{\text{train}} - T_{\text{fine}}}{\tau_2}\right) \quad (7)$$

Here, $\alpha$ acts as a "soft gate" that suppresses all geometric losses during the warm-up phase, effectively ensures that the VGGT teacher is only queried when the student model is capable of producing inputs. $\beta$ manages the transition from coarse camera pose constraint to fine-grained depth warping consistency.

Notably, the VGGT teacher and auxiliary losses are utilized only during training and entirely discarded at inference. This ensures that the model achieves superior geometric control without any additional computational overhead. Furthermore, our distillation strategy is architecture-agnostic; its versatility is demonstrated by the consistent performance gains across both UNet and Diffusion Transformer (DiT) backbones, as validated in our experiment.

## 4. Experiments

### 4.1. Experimental Setup

**Datasets, Evaluation Metrics, and Implementation Details.** We conduct training and evaluation on the standard RealEstate10K dataset (Zhou et al., 2018), and use the text descriptions provided by LAVIS (Li et al., 2023) as textual conditioning. For out-of-distribution testing, we employ the MannequinChallenge (Li et al., 2019), DL3DV (Ling et al., 2024) and ScanNet++ (Yeshwanth et al., 2023). We follow the previous works and adopt established metrics, including RotError (He et al., 2024), TransError (He et al., 2024), CamMC (Wang et al., 2024b), FVD (Unterthiner et al., 2018). More details, including Implementation Details, are shown in Appendix B.

**Comparison Methods.** We evaluate our approach using both U-Net and DiT architectures under sparse camera conditioning settings. For U-Net-based methods, we compare against CamI2V and CameraCtrl, although they are not for sparse conditioning, we adapt them to sparse conditioning via linear interpolation of camera poses. We exclude MotionCtrl (lacks image-to-video capability) and CamCo (code unavailable). While methods like SparseCtrl address sparse conditioning, they do not support explicit camera pose control and are therefore omitted. For DiT-based methods, sparse camera-conditioned image-to-video generation remains underexplored. Existing models (e.g., VD3D, CMG) lack image-to-video support, while other approaches (e.g., VACE, Wan) have not released camera pose conditioning yet. To establish a rigorous baseline, we fine-tune CogVideoX with a standard camera encoder.

We compare four configurations across both architectures: (1) Full: Models trained with dense camera supervision; (2) Full-interp: Full models tested with linearly interpolated pseudo-dense poses; (3) Base: Models trained directly on sparse camera poses; (4) CamGeo (Ours): Our proposed method with our distillation and curriculum learning.

### 4.2. Main Results

**Quantitative Results.** We conduct comprehensive quantitative evaluations under three sparsity settings (1/2, 1/3, and 1/4) on the RealEstate10K dataset. The results, summarized in Table 1, are categorized into UNet-based (Group I) and DiT-based (Group II) architectures.

*Performance on U-Net Architectures.* First, we analyze the limitations of geometric interpolation. As shown in Group I, standard interpolation-based baselines (CamI2V, CameraCtrl) exhibit significantly high errors, especially at the 1/4 sparsity setting. To investigate this further, we introduce the "SVD-Full-interp." baseline. Counter-intuitively, we observe that forcing the model to follow linearly interpolated poses yields *worse* performance than simply inferring on sparse inputs (SVD-Full). For instance, at the 1/4 setting, the Rotation Error of *SVD-Full-interp.* spikes to 1.72, significantly lagging behind *SVD-Full* (1.55). This critical finding suggests that rigid geometric interpolation introduces trajectory artifacts (e.g., unnatural constant velocity) that conflict with the complex, non-linear motion priors learned by the diffusion model. In contrast, our method consistently achieves the best performance. Notably, in the challenging 1/4 sparsity setting, our model surpasses SVD-Full by $\Delta 0.17$ and SVD-Base by $\Delta 0.25$ in Rotation Error, highlighting that our design effectively empowers the U-Net architecture to bridge sparse gaps using learned priors.

*Performance on DiT Architectures.* To verify the scalability of our approach, we extend our evaluation to the Diffusion Transformer (DiT) architecture in Group II. The results reveal two key insights. First, the limitation of interpolation is architecture-agnostic: similar to the U-Net findings, applying interpolation to DiT ("CogVideoX-Full-interp.") degrades performance compared to sparse inference (e.g., increasing RotError from 1.42 to 1.60 at 1/4 sparsity), further confirming that rigid geometric constraints hinder the model's generation process. Second, while the CogVideoX series demonstrates superior video generation quality thanks to the DiT backbone as proven by FVD, the base models struggle with precise camera alignment. CamGeo effectively addresses this, reducing the Rotation Error by 0.1 compared to the Full baseline while maintaining visual quality.

*Out-of-Distribution Evaluation.* In Table 1, we evaluate our approach on the MannequinChallenge dataset. Our method consistently outperforms baselines in both architecture groups, verifying its strong generalization capability under diverse and unseen scenes.

We further evaluate CamGeo on two additional benchmarks, DL3DV and ScanNet++, under multiple sparsity ratios (1/2, 1/3, and 1/4). We consider both zero-shot settings, where models trained on RealEstate10K are directly transferred to unseen domains, and in-domain training settings. As shown in Table (Ling et al., 2024), CamGeo consistently improves camera controllability and generation quality across both UNet-based (SVD) and DiT-based (CogVideoX) architectures, demonstrating strong architecture generalization.

Importantly, the performance gains remain stable as the

*Table 1.* Quantitative evaluation on RealEstate10K (in-distribution) and MannequinChallenge (out-of-distribution). Methods are split by backbone: Group I uses U-Net (e.g., SVD); Group II uses DiT (e.g., CogVideoX). "interp." denotes interpolation baselines. Results are shown for three sparsity ratios (1/2, 1/3, 1/4). Dark blue and light blue mark the best and second-best performance per group.

| Method | RealEstate10K (in-distribution) | | | | | MannequinChallenge (out-of-distribution) | | | | |
| | RotError ↓ | TransError ↓ | CamMC ↓ | FVD ↓ Style-GAN | FVD ↓ VideoGPT | RotError ↓ | TransError ↓ | CamMC ↓ | FVD ↓ Style-GAN | FVD ↓ VideoGPT |
| --- | --- | --- | --- | --- | --- | --- | --- | --- | --- | --- |
| *Sparsity Ratio (1/2)* | | | | | | | | | | |
| *Group I: U-Net Architectures* | | | | | | | | | | |
| CamI2V-interp. | 1.58 | 6.54 | 7.12 | 128.4 | 138.7 | 4.92 | 19.33 | 21.05 | 179.4 | 198.8 |
| CameraCtrl-interp. | 1.48 | 6.35 | 6.95 | 124.5 | 133.2 | 4.75 | 18.25 | 20.05 | 174.5 | 197.5 |
| SVD-Full-interp. | 1.54 | 6.45 | 7.05 | 126.8 | 136.5 | 4.82 | 18.45 | 20.25 | 176.8 | 199.5 |
| SVD-Full | 1.46 | 6.26 | 6.83 | 122.5 | 131.9 | 4.62 | 17.98 | 19.79 | 172.3 | 195.7 |
| SVD-Base | 1.55 | 5.13 | 5.85 | 105.9 | 116.1 | 3.45 | 13.21 | 16.11 | 155.3 | 168.2 |
| SVD-CamGeo (Ours) | 1.34 | 4.89 | 5.49 | 95.9 | 111.0 | 3.02 | 8.02 | 10.23 | 142.5 | 160.9 |
| *Group II: DiT Architectures* | | | | | | | | | | |
| CogVideoX-Full-interp. | 1.46 | 5.38 | 5.92 | 98.2 | 108.5 | 3.52 | 11.23 | 12.91 | 139.5 | 155.4 |
| CogVideoX-Full | 1.39 | 5.12 | 5.76 | 94.6 | 102.8 | 3.38 | 10.72 | 12.48 | 135.1 | 148.9 |
| CogVideoX-Base | 1.43 | 5.18 | 5.59 | 89.5 | 105.2 | 3.09 | 8.92 | 10.68 | 128.4 | 151.2 |
| CogVideoX-CamGeo (Ours) | 1.27 | 4.72 | 5.38 | 83.4 | 97.6 | 3.14 | 7.82 | 9.88 | 115.7 | 138.5 |
| *Sparsity Ratio (1/3)* | | | | | | | | | | |
| *Group I: U-Net Architectures* | | | | | | | | | | |
| CamI2V-interp. | 1.63 | 6.68 | 7.35 | 132.5 | 142.1 | 5.05 | 20.10 | 21.80 | 182.5 | 202.4 |
| CameraCtrl-interp. | 1.58 | 6.55 | 7.15 | 128.5 | 138.2 | 4.95 | 19.15 | 20.85 | 178.5 | 200.5 |
| SVD-Full-interp. | 1.62 | 6.65 | 7.25 | 130.5 | 140.8 | 4.98 | 19.45 | 21.15 | 180.2 | 202.5 |
| SVD-Full | 1.51 | 6.08 | 6.69 | 112.3 | 120.4 | 4.46 | 17.55 | 19.36 | 169.2 | 178.4 |
| SVD-Base | 1.58 | 5.05 | 5.80 | 97.1 | 116.9 | 3.38 | 14.93 | 15.83 | 149.0 | 170.5 |
| SVD-CamGeo (Ours) | 1.35 | 4.76 | 5.40 | 95.3 | 110.8 | 2.83 | 8.41 | 9.78 | 134.2 | 164.1 |
| *Group II: DiT Architectures* | | | | | | | | | | |
| CogVideoX-Full-interp. | 1.51 | 5.61 | 6.28 | 101.5 | 112.8 | 3.68 | 11.89 | 13.88 | 142.7 | 158.4 |
| CogVideoX-Full | 1.41 | 5.29 | 5.83 | 98.2 | 106.5 | 3.44 | 11.08 | 12.89 | 138.5 | 152.9 |
| CogVideoX-Base | 1.44 | 5.12 | 5.69 | 90.8 | 109.2 | 3.21 | 9.28 | 11.22 | 131.8 | 154.7 |
| CogVideoX-CamGeo (Ours) | 1.33 | 4.88 | 5.42 | 84.5 | 99.4 | 3.26 | 8.18 | 10.59 | 118.2 | 140.5 |
| *Sparsity Ratio (1/4)* | | | | | | | | | | |
| *Group I: U-Net Architectures* | | | | | | | | | | |
| CamI2V-interp. | 1.68 | 6.85 | 7.55 | 136.8 | 146.5 | 5.22 | 21.05 | 22.85 | 186.2 | 206.5 |
| CameraCtrl-interp. | 1.66 | 6.80 | 7.45 | 135.5 | 145.2 | 5.15 | 20.45 | 22.15 | 185.5 | 205.2 |
| SVD-Full-interp. | 1.72 | 6.85 | 7.50 | 134.5 | 148.5 | 5.10 | 20.25 | 22.05 | 188.5 | 208.2 |
| SVD-Full | 1.55 | 5.82 | 6.47 | 108.8 | 125.9 | 4.31 | 17.13 | 18.70 | 180.2 | 192.4 |
| SVD-Base | 1.63 | 4.95 | 5.72 | 95.6 | 118.9 | 3.25 | 14.78 | 15.59 | 152.4 | 174.2 |
| SVD-CamGeo (Ours) | 1.38 | 4.57 | 5.23 | 94.3 | 106.1 | 2.72 | 13.17 | 14.06 | 146.5 | 161.3 |
| *Group II: DiT Architectures* | | | | | | | | | | |
| CogVideoX-Full-interp. | 1.62 | 6.39 | 7.01 | 103.4 | 116.8 | 4.09 | 13.19 | 15.28 | 148.8 | 166.2 |
| CogVideoX-Full | 1.45 | 5.51 | 6.13 | 98.7 | 110.5 | 3.59 | 11.48 | 13.49 | 142.4 | 160.1 |
| CogVideoX-Base | 1.52 | 5.24 | 5.91 | 90.2 | 112.4 | 3.34 | 9.71 | 11.45 | 135.5 | 162.5 |
| CogVideoX-CamGeo (Ours) | 1.33 | 4.86 | 5.40 | 82.6 | 99.8 | 3.36 | 8.49 | 11.62 | 122.8 | 145.9 |

*Table 2.* User study results.

| Configuration | User Preference Rate ↑ | | Overall ↑ | Rank |
| | SVD-based | CogVideoX-based | | |
| --- | --- | --- | --- | --- |
| Full | 17.8 | 18.4 | 18.1 | 2 |
| Base | 9.7 | 11.7 | 10.7 | 3 |
| CamGeo | **72.5** | **69.9** | **71.2** | **1** |

sparsity ratio decreases from 1/2 to 1/4, indicating that CamGeo is robust to increasingly sparse camera supervision. In the zero-shot setting, CamGeo substantially reduces rotation and translation errors while improving motion consistency and perceptual video quality, suggesting superior out-of-domain generalization. Similar improvements are also observed under in-domain training.

**User Study.** We conducted a user study with 73 participants over 50 comparison groups sampled from three sparsity settings. Each comparison group was evaluated separately for the two backbone architectures. For the SVD-based setting, participants compared SVD-Full, SVD-Base, and SVD-CamGeo with the corresponding GT video as reference. For the CogVideoX-based setting, participants compared CogVideoX-Full, CogVideoX-Base, and CogVideoX-CamGeo with the corresponding GT video as reference. The GT video was used only to indicate the target camera trajectory and was not included as a candidate method.

To ensure unbiased judgment, we anonymized all methods and settings and randomly shuffled the video order. Participants were asked to select the result that best aligned with the GT camera trajectory while maintaining high visual smoothness. As shown in Table 2, CamGeo is preferred in 71.2% of all comparisons. These results indicate that our method provides better trajectory alignment, temporal consistency, and camera-conditioned motion control.

### 4.3. Ablation Study

Table 4 contains ablation study results of our components in different architectures and sparsity.

**Impact of Smoothness Mechanism.** Table 4a shows that removing the smoothness mechanism leads to a clear performance drop in all camera-related metrics (e.g., RotError increases from 1.34 to 1.45). This confirms that effective inter-frame interaction is critical for maintaining temporal coherence when camera conditions are sparse.

*Table 3.* Performance of on DL3DV and ScanNet++.

| Dataset | Sparsity | Setting | Method | RotError↓ | TransError↓ | CamMC↓ | FVD↓ Style-GAN | FVD↓ Video-GPT |
|---|---|---|---|---|---|---|---|---|
| **DL3DV** (U-Net) | 1/2 | Zero-shot (OOD) | SVD-Full | 3.58 | 13.42 | 15.31 | 158.2 | 180.5 |
| | | | SVD-Base | 3.92 | 14.81 | 17.26 | 165.4 | 186.2 |
| | | | SVD-CamGeo | **3.06** | **10.68** | **12.19** | **148.9** | **170.6** |
| | | In-Domain Trained | SVD-Full | 3.44 | 12.63 | 14.31 | 154.5 | 177.8 |
| | | | SVD-Base | 2.81 | 10.14 | 11.52 | 142.8 | 161.9 |
| | | | SVD-CamGeo | **2.04** | **7.49** | **8.76** | **126.3** | **144.5** |
| | 1/3 | Zero-shot (OOD) | SVD-Full | 3.89 | 15.23 | 17.02 | 166.1 | 187.4 |
| | | | SVD-Base | 4.16 | 16.34 | 18.63 | 172.8 | 192.1 |
| | | | SVD-CamGeo | **3.34** | **13.08** | **14.47** | **153.6** | **176.3** |
| | | In-Domain Trained | SVD-Full | 3.73 | 14.54 | 16.21 | 163.9 | 181.6 |
| | | | SVD-Base | 3.11 | 11.85 | 13.14 | 148.7 | 168.4 |
| | | | SVD-CamGeo | **2.33** | **8.62** | **10.04** | **132.1** | **152.6** |
| | 1/4 | Zero-shot (OOD) | SVD-Full | 4.31 | 17.42 | 19.43 | 172.6 | 194.1 |
| | | | SVD-Base | 4.58 | 18.72 | 20.54 | 177.5 | 197.3 |
| | | | SVD-CamGeo | **3.64** | **14.78** | **16.42** | **161.9** | **184.2** |
| | | In-Domain Trained | SVD-Full | 4.12 | 16.85 | 18.52 | 170.6 | 193.1 |
| | | | SVD-Base | 3.42 | 13.51 | 15.14 | 158.4 | 180.5 |
| | | | SVD-CamGeo | **2.68** | **9.81** | **11.37** | **138.2** | **161.4** |
| **ScanNet++** (DiT) | 1/2 | Zero-shot (OOD) | CogVideoX-Full | 1.61 | 5.96 | 6.62 | 100.8 | 114.7 |
| | | | CogVideoX-Base | 1.68 | 6.24 | 6.87 | 103.4 | 116.3 |
| | | | CogVideoX-CamGeo | **1.51** | **5.62** | **6.34** | **95.7** | **111.4** |
| | | In-Domain Trained | CogVideoX-Full | 1.54 | 5.86 | 6.48 | 102.6 | 114.2 |
| | | | CogVideoX-Base | 1.41 | 5.43 | 5.98 | 91.8 | 107.5 |
| | | | CogVideoX-CamGeo | **1.26** | **4.74** | **5.34** | **86.4** | **98.7** |
| | 1/3 | Zero-shot (OOD) | CogVideoX-Full | 1.76 | 6.43 | 7.08 | 106.4 | 121.5 |
| | | | CogVideoX-Base | 1.84 | 6.72 | 7.31 | 109.2 | 124.6 |
| | | | CogVideoX-CamGeo | **1.63** | **6.04** | **6.65** | **100.3** | **116.8** |
| | | In-Domain Trained | CogVideoX-Full | 1.72 | 6.44 | 7.02 | 108.9 | 120.7 |
| | | | CogVideoX-Base | 1.55 | 5.68 | 6.24 | 97.4 | 112.5 |
| | | | CogVideoX-CamGeo | **1.38** | **4.97** | **5.58** | **90.3** | **104.2** |
| | 1/4 | Zero-shot (OOD) | CogVideoX-Full | 1.94 | 6.94 | 7.62 | 112.1 | 126.7 |
| | | | CogVideoX-Base | 2.05 | 7.23 | 7.85 | 116.4 | 128.5 |
| | | | CogVideoX-CamGeo | **1.82** | **6.35** | **7.04** | **107.6** | **122.9** |
| | | In-Domain Trained | CogVideoX-Full | 1.96 | 6.91 | 7.52 | 116.1 | 128.8 |
| | | | CogVideoX-Base | 1.76 | 6.21 | 6.84 | 103.7 | 118.4 |
| | | | CogVideoX-CamGeo | **1.54** | **5.41** | **5.98** | **93.5** | **110.1** |

*Table 4.* Ablation studies on smoothness, warm-up, curriculum scheduling, modalities, and weighting strategies.

*(a)* Cross-Frame Smoothness (1/2 sparse)

| SVD | w/o Smoothness | Ours |
|---|---|---|
| RotError ↓ | 1.45 | 1.34 |
| TransError ↓ | 5.05 | 4.89 |
| CamMC ↓ | 5.71 | 5.49 |
| FVD-StyleGAN ↓ | 103.5 | 95.9 |
| FVD-VideoGPT ↓ | 116.6 | 111.0 |

*(b)* Warm-up (1/3 sparse)

| SVD | w/o Warm-up | Ours |
|---|---|---|
| RotError ↓ | 1.48 | 1.35 |
| TransError ↓ | 5.12 | 4.76 |
| CamMC ↓ | 5.83 | 5.40 |
| FVD-StyleGAN ↓ | 99.2 | 95.3 |
| FVD-VideoGPT ↓ | 114.7 | 110.8 |

*(c)* Long-range smooth (1/4 sparse)

| SVD | Only Adjacent | Ours |
|---|---|---|
| RotError ↓ | 1.43 | 1.38 |
| TransError ↓ | 5.61 | 4.57 |
| CamMC ↓ | 6.20 | 5.23 |
| FVD-StyleGAN ↓ | 112.5 | 94.27 |
| FVD-VideoGPT ↓ | 125.4 | 106.1 |

*(d)* Curriculum scheduling (1/2 sparse)

| CogVideoX | Linear | Ours |
|---|---|---|
| RotError ↓ | 1.33 | 1.27 |
| TransError ↓ | 4.86 | 4.89 |
| CamMC ↓ | 5.53 | 5.38 |
| FVD-StyleGAN ↓ | 86.4 | 83.4 |
| FVD-VideoGPT ↓ | 101.8 | 97.6 |

*(e)* Two modalities (1/3 sparse)

| CogVideoX | Camera | Depth | Ours |
|---|---|---|---|
| RotError ↓ | 1.39 | 1.43 | 1.33 |
| TransError ↓ | 5.52 | 5.19 | 4.88 |
| CamMC ↓ | 6.04 | 5.74 | 5.42 |
| FVD-StyleGAN ↓ | 105.1 | 99.8 | 84.5 |
| FVD-VideoGPT ↓ | 101.4 | 103.6 | 99.4 |

*(f)* Dynamic weight (1/4 sparse)

| CogVideoX | w/o Dynamic | Ours |
|---|---|---|
| RotError ↓ | 1.41 | 1.33 |
| TransError ↓ | 5.03 | 4.86 |
| CamMC ↓ | 5.61 | 5.40 |
| FVD-StyleGAN ↓ | 87.7 | 82.6 |
| FVD-VideoGPT ↓ | 107.7 | 99.8 |

**Impact of Warm-Up.** Tab 4b reveals that the absence of a warm-up phase causes significant degradation across all key indicators, with CamMC rising from 5.40 to 5.83. This underscores the necessity of the warm-up stage in stabilizing the initial generation process under sparse conditions.

**Impact of Long-Range Smoothness.** Tab 4c shows that relying only on adjacent-frame smoothness substantially increases TransError (4.57 to 5.61). This underscores the necessity of the warm-up stage in stabilizing the initial generation process under sparse conditions.

**Impact of Curriculum Scheduling.** Table 4d compares our sigmoidal schedule with linear interpolation. Our approach achieves superior RotError (1.27) and CamMC (5.38), indicating that a proper curriculum transition provides a more effective learning trajectory for complex camera motions.

**Impact of Two Modalities.** We evaluate the synergy of camera pose and depth in Table 4e. Using either modality alone results in higher error metrics (e.g., TransError>5.1; only their combined use achieves the best geometric accuracy, proving that both modalities provide complementary

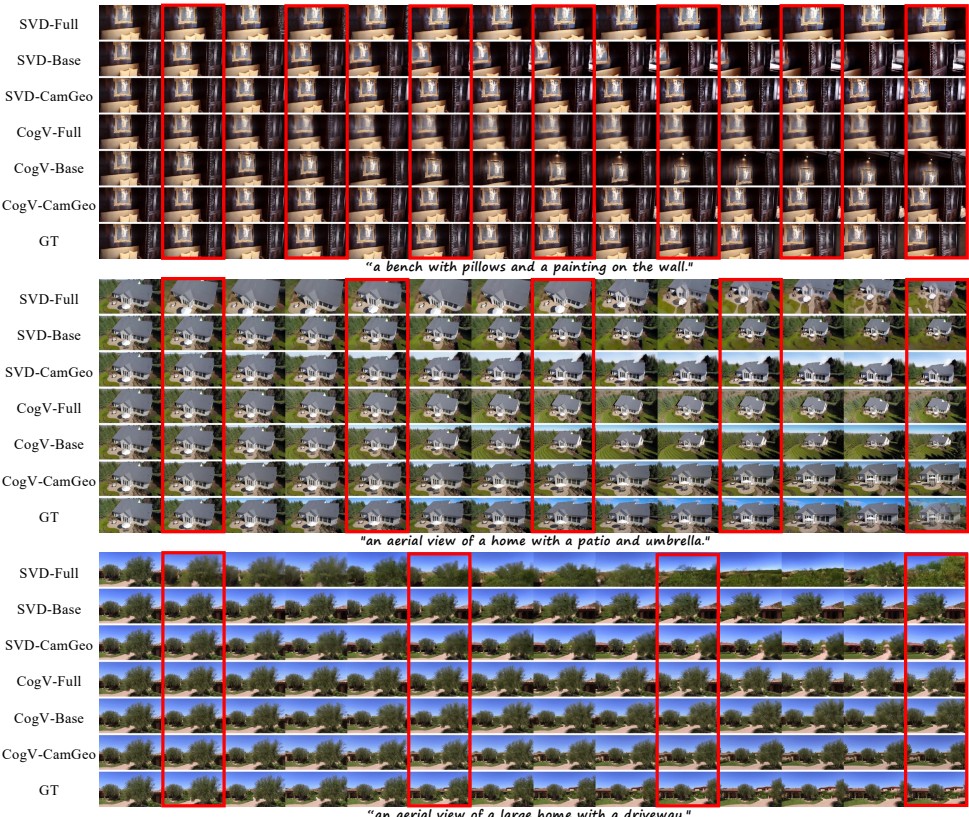

*Figure 4.* Qualitative results on RealEstate10K. Red boxes highlight frames with camera conditioning; "CogV" denotes CogVideoX.

benefits.

**Impact of Dynamic Weight.** Tab 4f demonstrates that replacing our dynamic weighting with a constant alternative increases all error metrics, such as RotError rising to 1.41. This confirms dynamic weighting is crucial for optimizing information flow from conditional to unconditional frames.

**Qualitative Results.** To further demonstrate the effectiveness of our approach, we provide qualitative comparisons in Fig. 4. **1) Camera trajectory adherence.** CamGeo achieves superior camera following capabilities and generalizes well across different backbone architectures (e.g., UNet and DiT). As shown in the 1st row, the camera is intended to turn left toward the closet; in the 2nd row, the viewpoint descends relative to the house; and in the 3rd row, the camera pans to the left. In all three scenarios, only Cam-Geo successfully executes the target motions, whereas the baseline models frequently fail to align with the guidance. These results suggest that our method effectively decouples camera control from content generation, allowing it to override the inherent motion biases of the pre-trained models. **2) Geometric realism.** In addition to precise motion control, our method maintains more stable geometry under dynamic viewpoint changes. For instance, in the comparison within the 1st row, the paintings hanging on the wall produced by

SVD-Full, SVD-Base, and CogVideoX-Full appear significantly blurred or indistinct. By contrast, CamGeo not only adheres better to the camera trajectory but also effectively preserves the general contours and geometric details of the paintings. This discrepancy highlights a critical limitation in current video generation models: the trade-off between motion magnitude and texture fidelity. The blurred artifacts in the baselines indicate a loss of structural coherence during spatial transformations. Conversely, our results demonstrate that by explicitly integrating geometric constraints, Cam-Geo can prevent the texture collapse. Besides, **Failure Case Analysis** can be viewed in Appendix D.

## 5. Conclusion

We presented CamGeo, a novel framework for sparse camera-conditioned image-to-video generation. To address the critical challenges of pose drift and jerky motion under sparse supervision, CamGeo leverages rich 3D geometric priors from VGGT through a training-only distillation strategy. Furthermore, a coarse-to-fine curriculum learning strategy ensures stable training by progressively scaling geometric complexity. Extensive experiments validate that CamGeo achieves consistent improvements across various sparsity settings.

## Acknowledgements

This work was supported by the National Key R&D Program of China No. 2024YFB2809103, National Science Foundation of China No. U25B2010 and Beijing Natural Science Foundation No. L242014, and New Cornerstone Science Foundation through the XPLORER PRIZE.

## Impact Statement

This paper presents work whose goal is to advance the field of machine learning. There are many potential societal consequences of our work, none of which we feel must be specifically highlighted here.

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

# Appendices

The Appendix is organized as follows:

Section A: provides more details of the training diffusion model's coarse-to-fine process.

Section B: provides more details about dataset, evaluation metrcis, and implementation(including training and hyper-parameters).

Section C: provides more qualitative examples of in-distribution and out-of-distribution evaluation.

Section D: gives analysis of failure cases.

Section E: provides analysis on the limitation.

In the supplementary material, we provide 10 representative comparison videos, and we highly encourage readers to view them.

## A. Empirical Observation of Training process.

Based on the qualitative results shown in Fig 5, we observe a clear coarse-to-fine evolution during training of image-to-video diffusion model. We obtain the decoded frames $\hat{V}$ and find that: in the early training stages, $\hat{V}$ exhibit substantial blur and pronounced geometric inconsistencies. For instance, in the first example (1st row), the gate post becomes severely blurred, the overall structural integrity collapses, and the estimated camera pose deviates significantly from the ground truth. As training proceeds, the model gradually stabilizes: structures become sharper and more coherent, and the predicted camera trajectories increasingly resemble those of the GT. By the 5th row, the geometric layout is considerably more faithful, and fine structural boundaries begin to emerge. A similar trend is evident in the fourth example: within the green dashed box, the pillow transitions from a vague, blurry blob to a sharply defined object with pixel-level correspondence to the GT. Likewise, in the orange dashed box, the red pillow's position—initially misaligned—progressively converges toward the correct GT location as iterations increase. Comparable coarse-to-fine improvements consistently appear across other examples as well.

These observations indicate that, at early training stages, the model struggles to establish coherent spatial relationships and object geometries, largely due to the inherent difficulty of simultaneously learning global scene structure and fine-grained details from scratch within a high-dimensional diffusion process.

To mitigate this learning challenge, we adopt a curriculum learning strategy that adaptively adjusts supervisory granularity over the course of training. After warm-up, in the initial phase, we introduce camera pose as a coarse-grained conditioning signal, enabling the model to first internalize global scene layout and geometric constraints without being overwhelmed by local details. As the model's outputs become sharper and more structurally stable, we gradually transition to depth maps as fine-grained, pixel-level supervision. This staged refinement allows the model to progressively enhance local geometric fidelity, improve structural alignment, and ultimately produce more photorealistic and geometrically consistent video sequences. Such a coarse-to-fine training paradigm not only stabilizes optimization but also aligns naturally with the intrinsic learning dynamics observed in diffusion models.

## B. More Details about Dataset, Evaluation Metrcis, and Implementation

### B.1. IMPLEMENTATION DETAILS

**Model selection.** For UNet architecture, we build upon a pre-trained Stable Video Diffusion (SVD) (Blattmann et al., 2023) model. We integrate a lightweight pose encoder on top of SVD to process sparse camera inputs. Similarily, our implementation is built on CogVideoX, the VAE's temporal compression ratio is set to 1 to adapt to our sparse conditions.

**Text prompt.** As for the text prompt of RealEstate10K Dataset, we using the prompts offered by CameraCtrl (He et al., 2024), which leverages LAVIS (Li et al., 2023) to generate a caption for each video clip. For MannequinChallenge Dataset, we use BLIP-2 (Li et al., 2022).

**Hyper-parameters.** We set $\lambda^{(1)} = 1, \lambda^{(2)} = 1, \lambda^{(3)} = 1$ to encourage adjacent and long-range smoothness if applicable. For curriculum scheduling, we set $T_{\text{warm}} = 200$ and $T_{\text{fine}} = 600$. And the Sigmoid scheduling hyper-parameters are set to $\tau_1 = 40$ and $\tau_2 = 80$.

**Optimization.** Adam (Kingma, 2014) optimizer is leveraged to train our model with a constant learning rate of $3 \times 10^{-5}$ and trained on $16\times$ NVIDIA H20 GPUs for 16K steps with a batch size of 16.

### B.2. EVALUATION METRICS

To mitigate randomness introduced by COLMAP, we run five independent trials for each of the 1,000 sampled videos and compute the average using only the trials that successfully complete for that sample.

**RotError** (He et al., 2024). We evaluate per-frame camera-to-world rotation accuracy by the relative angles between ground truth rotations $R_i$ and estimated rotations $\tilde{R}_i$ of generated frames. We report accumulated rotation error along all frames in radians.

$$\text{RotErr} = \sum_{i=1}^{n} \cos^{-1} \frac{\text{tr}\left(\tilde{R}_i R_i^{\text{T}}\right) - 1}{2} \qquad (8)$$

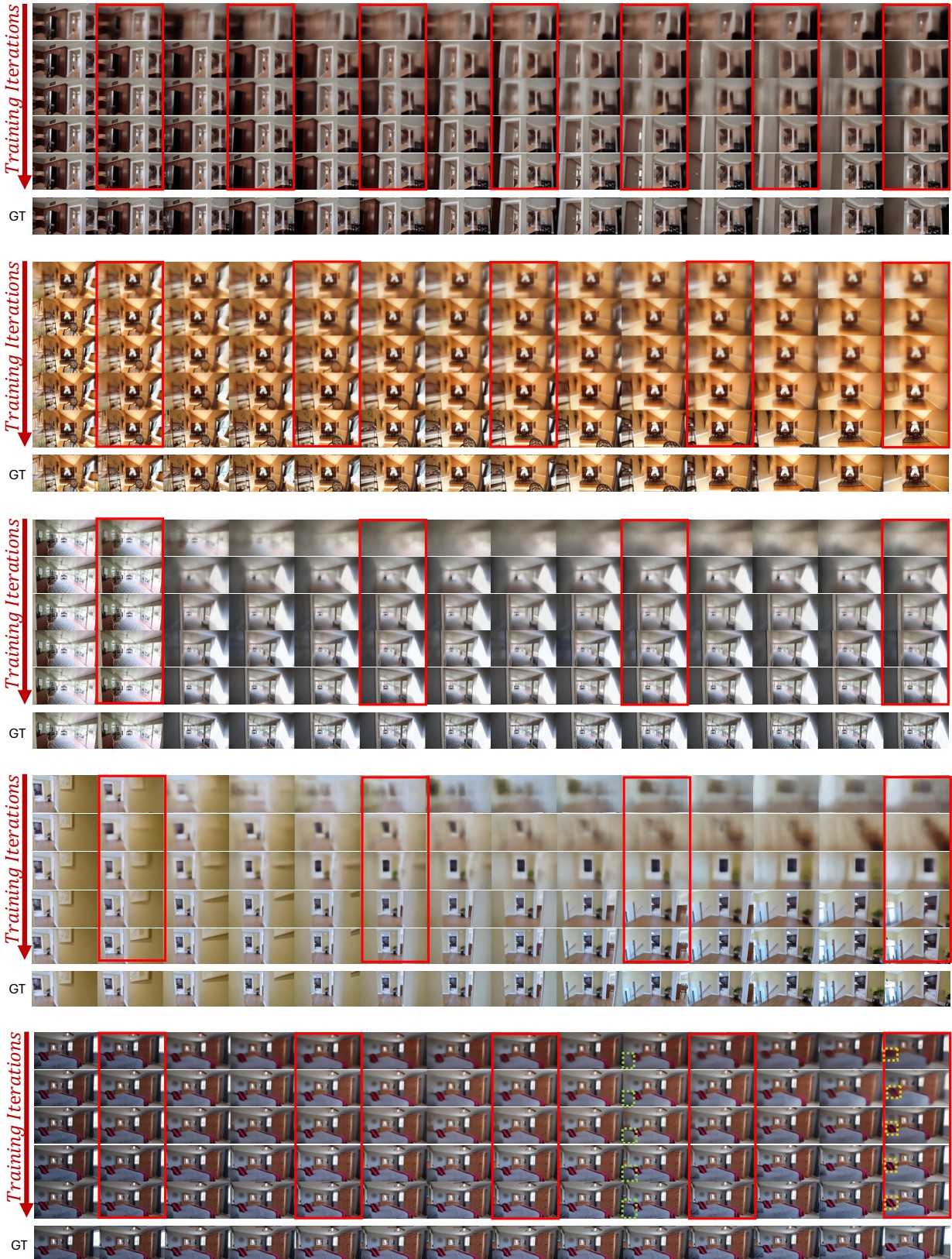

*Figure 5.* The training coarse-to-fine process. At early training iterations, comparing with GT, the training samples are geometric inconsistent, while in later iterations. Red boxes indicate frames with camera conditioning.

**TransError** (He et al., 2024). We evaluate per-frame camera trajectory accuracy by the camera location in the world coordinate system, i.e. the translation component of camera-to-world matrices. We report the sum of $\mathcal{L}_\in$ distance between ground truth translations $T_i$ and generated translations $\tilde{T}_i$ for all frames:

$$\text{TransErr} = \sum_{i=1}^{n} \left\| \tilde{T}_i - T_i \right\|_2 \qquad (9)$$

**CamMC** (Wang et al., 2024b). We also evaluate camera pose accuracy by directly calculating L2 similarity of per-frame rotations and translations as a whole. We sum up the results of all frames.

$$\text{CamMC} = \sum_{i=1}^{n} \left\| \left[ \tilde{R}_i | \tilde{T}_i \right] - [R_i | T_i] \right\|_2 \qquad (10)$$

**FVD (Unterthiner et al., 2018).** To ensure that our proposed method coherently enhances the generative capability and visual quality of the base I2V model, we employ Fréchet Video Distance (FVD) to measure the distance between the generated frames and the training data distribution by Style-GAN (Karras et al., 2019) and VideoGPT (Yan et al., 2021).

## C. More Experimental Results

### C.1. IN-DISTRIBUTION RESULTS

In Fig. 6, to further validate the effectiveness of our approach, we present more qualitative comparisons. The qualitative results present great performance of our method from three perspectives:

**1) Camera trajectory adherence.** In the 2nd row, Cam-Geo (both UNet and DiT) moves towards near the blue bed, while other comparisons failt to follow the GT trajectories. Similarly, in the 3rd row, the cameras of SVD-Full and SVD-Base moves forward, while CamGeo follows GT's camera moves backward. These results indicate that our sparse camera-conditioning module successfully constrains viewpoint transitions, producing smooth, stable, and physically plausible trajectories even under sparse supervision.

**2) Geometric realism.** Under sparse conditioning, baseline methods often exhibit blur, texture degradation, or geometric distortions. For instance, in the 1st row, SVD-Base produces warped buildings, and in the 4th row, the orange floor becomes noticeably blurred. In contrast, our approach consistently preserves fine-grained geometry across frames. These comparisons demonstrate that the integration of 3D priors enables our model to maintain structural accuracy and recover coherent scene geometry even when camera observations are sparse.

Overall, these qualitative results highlight that our method delivers substantial improvements in trajectory fidelity, ge-ometric quality and coherence compared to baseline approaches.

### C.2. OUT-OF-DISTRIBUTION RESULTS

To further assess the generalization ability of our approach, we conduct additional evaluations on the out-of-distribution MannequinChallenge dataset (Li et al., 2019), as shown in Fig. 7. Despite the significant domain shift, our method consistently demonstrates superior camera-trajectory coherence, geometric fidelity compared to existing baselines. In the 1st row, SVD-Full fails to maintain plausible geometry—the hairdresser's head collapses noticeably, while SVD-Base exhibits a large, unstable camera drift. In contrast, SVD-CamGeo preserves a stable trajectory that remains well aligned with the GT sequence. In the 2rd row, the photographer rendered by SVD-Full and SVD-Base collapses, revealing its inability to infer consistent object motion under distribution shift. At the same time, CogVideoX-Full and CogVideoX-Base's moving direction does not math with GT, while CogVideoX-Geo aligns with GT.

These results collectively demonstrate that our framework not only performs well in-distribution but also exhibits strong robustness when deployed in challenging, real-world scenes.

## D. Failure Case Study

Figure 8 illustrates several representative failure cases of our method.

First, in scenarios with extremely rapid camera transitions (Row 1), the model occasionally produces motion blur and spatial-temporal inconsistencies, such as the blurring of the table and sofa.

Second, when the viewpoint transitions into a previously unobserved room (Row 2), the model struggles with disocclusion, leading to content collapse in newly revealed regions. This indicates that the diffusion prior lacks sufficient inpainting capability for large-scale scene transitions without additional guidance.

Third, as shown in Row 3, shadows often remain static despite changing viewpoints, reflecting the model's failure to capture view-dependent lighting effects and its tendency to treat shadows as fixed textures rather than dynamic optical phenomena.

## E. Limitation

Although our approach substantially improves camera-pose consistency and long-range coherence under sparse conditioning, it still inherits limitations of diffusion-based video generation. In particular, extending the framework to produce very long video sequences remains challenging, as

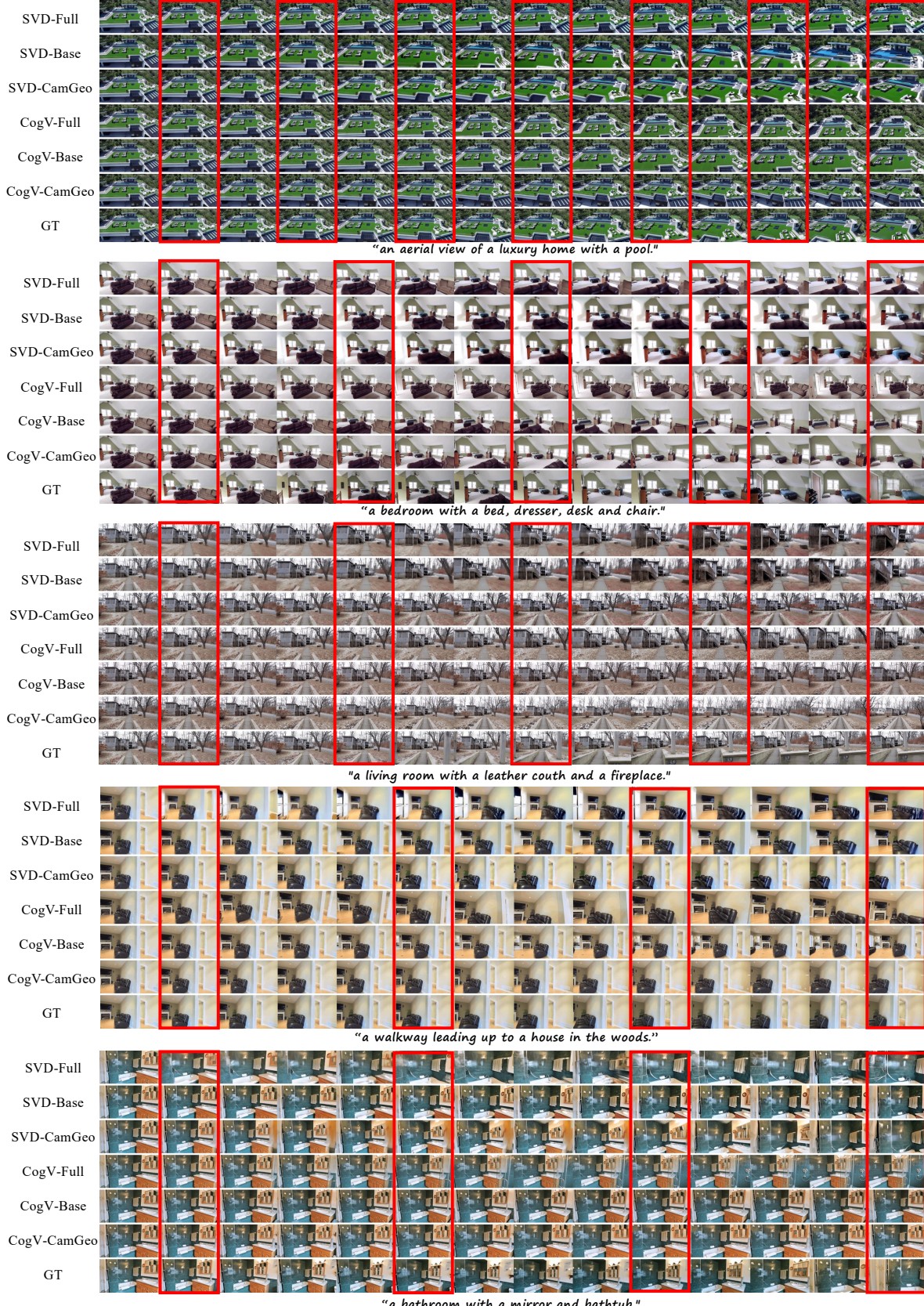

*Figure 6.* More qualitative results on RealEstate10K. Red boxes indicate camera-conditioned frames.

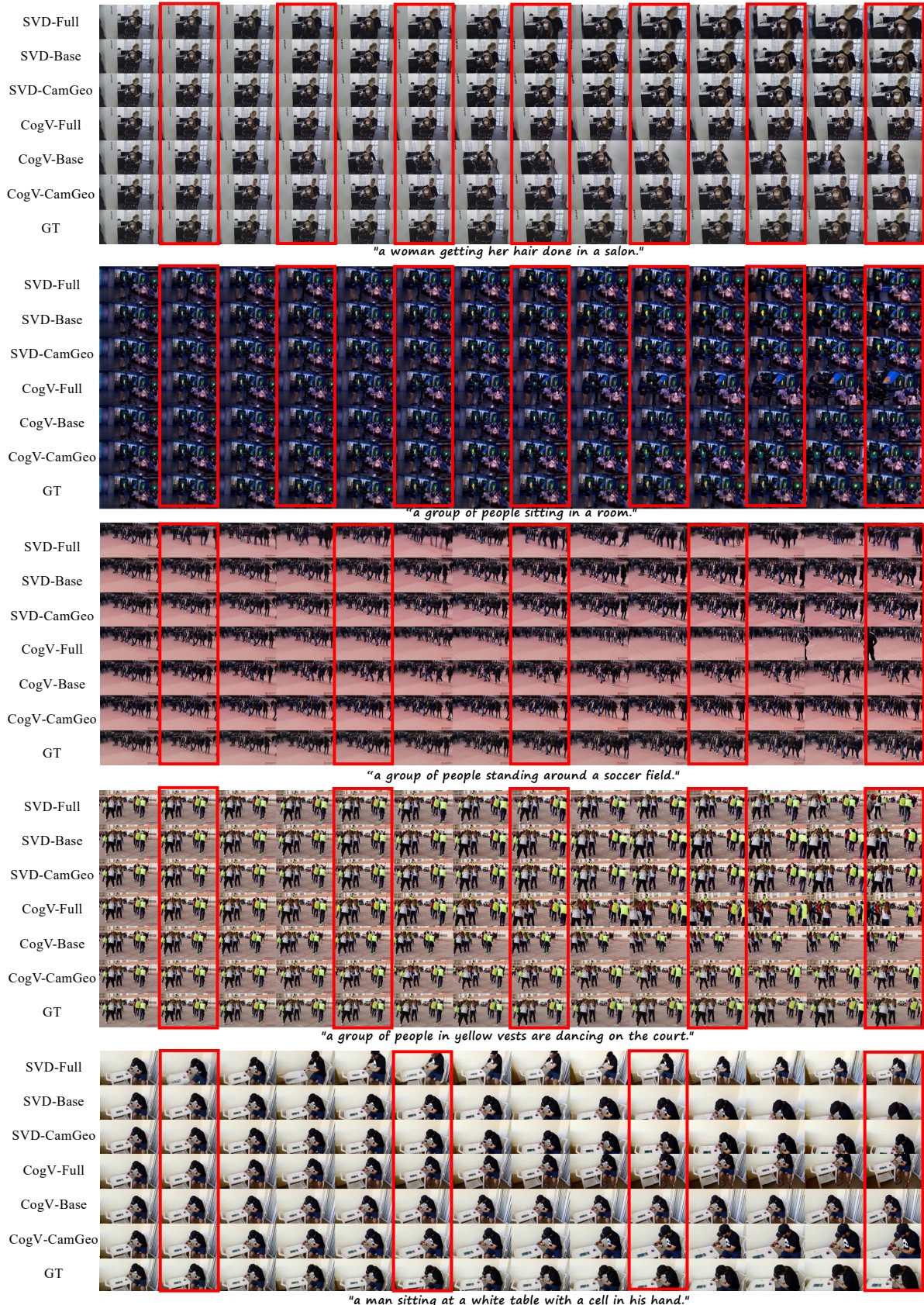

*Figure 7.* Out-of-distribution results on MannequinChallenge. Red boxes indicate camera-conditioned frames.

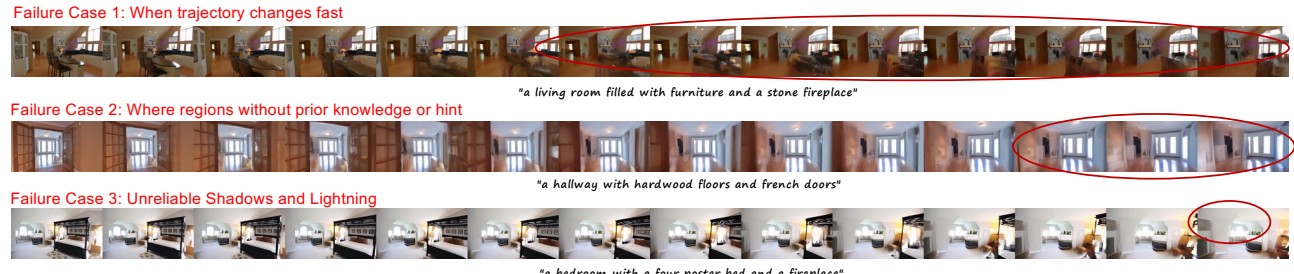

*Figure 8.* Failure Case Analysis. Red circles denote the failure regions.

temporal drift can accumulate beyond the temporal window modeled by our 3D-guided architecture. Moreover, our design focuses on leveraging sparse camera cues within a fixed-range generation process, which may limit its ability to maintain global structure over extremely long horizons. Developing more scalable temporal modeling strategies or memory-augmented architectures is an exciting direction for future work.

