# OpenReview forum: "CamGeo: Sparse Camera-Conditioned Image-to-Video Generation with 3D Geometry Priors"
_ICML.cc/2026/Conference — ICML 2026 regular_

### Official Review · Reviewer_AvVF · 2026-03-12

**Soundness:** 3
**Presentation:** 3
**Significance:** 2
**Originality:** 3
**Overall Recommendation:** 4
**Confidence:** 3

**Summary:**

This paper studies sparse camera-conditioned image-to-video generation, where camera poses are provided only for a subset of keyframes rather than for every frame. The proposed method, CamGeo, uses a frozen VGGT model as a training-time geometry teacher and removes it at inference time, so inference cost remains unchanged. The method combines three ingredients: (1) keyframe trajectory distillation to align generated keyframes with the provided sparse poses, (2) cross-frame geometric consistency distillation based on teacher-estimated camera and depth signals to regularize intermediate frames, and (3) a coarse-to-fine curriculum to stabilize optimization. Experiments are conducted across multiple sparsity ratios, two backbone families, and both in-distribution and out-of-distribution settings.

**Compliance With Llm Reviewing Policy:**

Affirmed.

**Final Justification:**

I admire the authors' efforts for the detailed rebuttal, and my main concerns are addressed. I will keep my positive rating.

**Key Questions For Authors:**

- Since dense-trajectory datasets such as RealEstate10K are available, can the authors clarify what is gained by the proposed sparse-only training framing beyond showing that geometry distillation improves sparse-conditioned fine-tuning?

- What is the actual training-time overhead of using the frozen VGGT teacher, in terms of wall-clock time and GPU memory, relative to the Base model?

**Limitations:**

yes

**Strengths And Weaknesses:**

**Strengths**

- The core idea is sensible. Distilling geometric priors from a strong 3D teacher during training while keeping inference-time cost unchanged is a well-motivated design.

- The method is coherent. The keyframe anchoring term, cross-frame geometric regularization, and curriculum learning form a reasonably unified training recipe.

- The empirical study is fairly broad, including multiple sparsity ratios, two backbone families, out-of-distribution evaluation, ablations, and a user study. The most convincing empirical result is the consistent gain over the sparse-conditioning base models, suggesting that geometry-aware training adds value beyond sparse fine-tuning alone.


**Weaknesses**

- The paper overstates the necessity of sparse-train / sparse-test alignment. The experiments show that geometry distillation helps under sparse conditioning, but they do not fully establish that dense-data-based training is an inadequate starting point.

- The comparison set is useful but not fully decisive. The paper includes interpolation baselines and internal variants, but stronger sparse-control baselines would make the empirical case more convincing.

- The sparse controls are obtained by subsampling dense trajectories, which is a reasonable benchmark protocol but still somewhat idealized. Robustness to noisy or user-specified sparse camera inputs is not tested.

- The paper emphasizes “no inference overhead,” but does not clearly quantify the additional training-time cost of the frozen VGGT teacher in wall-clock time and GPU memory.

---

> ### Author Rebuttal · Authors · 2026-03-31
>
> We thank the reviewer for the constructive feedback and the positive assessment of our work. Below, we provide detailed responses to the specific comments and clarify the core technical contributions of CamGeo.
>
> ### For Weakness 1 & Question 1
> This is a thoughtful perspective. Indeed, dense-pose training is a strong starting point; however, its practical scalability is severely limited. While datasets like RealEstate10K provide dense poses, such ideal conditions are rare in real-world scenarios. Scaling models to millions or billions of in-the-wild internet videos would require an almost impossible amount of dense annotation, the cost of which is prohibitive.
>
> The ultimate value of CamGeo lies in using sparse-conditioned training as a starting point. By leveraging the VGGT teacher to compensate for missing geometric conditions, CamGeo establishes a highly practical and scalable technical path for training 3D video generation models on massive, weakly-annotated data. Furthermore, even if dense data were available, training on sparse conditions remains essential to mitigate the distribution shift between training and inference, as real-world testing typically involves sparse inputs. We will revise the paper to emphasize CamGeo’s role in enabling large-scale, practical 3D video training.
>
> ### For Weakness 2
> We appreciate your suggestion. It is important to clarify that, existing camera-conditioned image-to-video methods have not addressed the "sparse-train/sparse-test" regime, which is a highly practical yet challenging scenario. Precisely for this reason, our work is the first to focus on this setting, proposing CamGeo to solve the critical issue of insufficient camera signals under sparse conditioning. Consequently, there are no off-the-shelf existing methods specifically designed for this task. To demonstrate the superiority of our approach, we conducted extensive comparative experiments by adapting the strongest current base models using pose interpolation and multiple internal variants. This ensures a rigorous and fair evaluation against the most advanced camera-control frameworks currently available.
>
> ### For  Weakness 3
> This is indeed a highly valuable point. In fact, CamGeo is inherently resilient to imperfect inputs because it is trained on real-world datasets (e.g., RealEstate10K) that naturally contain diverse camera trajectories and inherent SfM estimation noises. This extensive exposure allows the model to learn a robust mapping that prioritizes global 3D geometric structure over local coordinate jitter.
>
> Furthermore, to quantitatively evaluate this robustness, we conduct a Perturbation Stress Test by adding $\pm $  10% random uniform perturbation to the user-specified sparse input poses (both Rotation and Translation). We measure the resulting performance against the clean Ground Truth. As shown in the **Table** below, the performance of SVD-CamGeo remains exceptionally stable under perturbation. Output errors increase only minimally (e.g., RotError shifts slightly from 1.38 to 1.43, and TransError stays well under 5.0), while visual quality metrics (FVD) show almost no degradation. These results demonstrate the high robustness of our method.
>
> | Evaluation Setting | Method | RotError ↓ | TransError ↓ | CamMC ↓ | FVD-Style-GAN ↓  | FVD-VideoGPT ↓  |
> | :--- | :--- | :--- | :--- | :--- | :--- | :--- |
> | Clean Input (Upper Bound) | SVD-CamGeo (Ours) | 1.38 | 4.57 | 5.23 | 94.3 | 106.1 |
> | Perturbation Input (±10% Noise) | SVD-CamGeo (Ours) | 1.43 | 4.78 | 5.35 | 95.9 | 107.2 |
>
> ### For Weakness 4 & Question 2
> In our experiments, for SVD, CamGeo requires 21 hours of training and 59.51 GB of GPU memory, compared with 16 hours and 50.14 GB for the base model. For CogVideoX, CamGeo takes 34 hours to train with 80.27 GB of GPU memory usage, whereas the base model requires 27 hours and 69.28 GB. With the introduction of VGGT supervision, CamGeo incurs a reasonable increase in training time and a marginal addition in memory usage, both of which remain well within a manageable range of computational resources. Notably, this VGGT supervision is active only during training and is removed during inference, thus introducing zero additional inference overhead. We add these details to the revised manuscript for better reproducibility.
>
> We sincerely thank the reviewer again for these insightful comments. We will incorporate all the aforementioned clarifications and additional results into the revised version to further strengthen the paper.

---

> > ### Author Rebuttal · Reviewer_AvVF · 2026-04-03
> >
> > I would like to thank the authors for their detailed and constructive rebuttal. The newly added perturbation stress test (W3) and the transparent breakdown of the training overhead (W4/Q2) are much appreciated. These additions effectively address my concerns regarding robustness and efficiency.
> >
> > However, I still have some reservations regarding the core justification for the sparse-training paradigm (W1/Q1). The rebuttal argues that obtaining dense annotations for large-scale data is "prohibitive." Yet, the proposed method relies on a frozen VGGT model to extract dense pseudo-labels (camera trajectories and depth) on the fly during training. If the VGGT teacher is efficient enough to provide these dense signals during the training loop, it could presumably be used to pre-process dense annotations offline. This makes the argument that "dense training is impractical" somewhat contradict the method's own capability.
> >
> > While this logical gap in the motivational framing remains, I acknowledge that the geometry distillation technique itself is technically solid, and the empirical improvements over the sparse-conditioned base models are clear.
> >
> > Therefore, I will maintain my original score of 4 (Weak Accept). I encourage the authors to refine their motivational claims in the final manuscript to better reflect the true bottleneck they are addressing.

---

> > > ### Author Response · Authors · 2026-04-04
> > >
> > > We would like to express our sincere gratitude to you for the insightful follow-up comments and for acknowledging our newly added stress tests and overhead analysis. We are particularly grateful for the highly inspiring suggestion regarding offline dense-annotation pre-processing using VGGT.
> > >
> > > Meanwhile, we would like to provide further clarification. While VGGT could be used to pre-process dense pseudo-labels offline, the other key motivation of CamGeo is to maintain train-test consistency and address the inherent distribution shift. Since the model is expected to generate full videos from sparse camera inputs during inference, we keep the inputs sparse during training as well. Providing dense pseudo-camera poses as inputs during training (even if they are easily obtainable via VGGT) risks biasing the model to rely on frame-by-frame guidance (a "dense-input dependency"). This would break the train-test consistency, potentially hindering the model's ability to hallucinate or internalize missing geometry when presented with only sparse conditions during inference.
> > >
> > > Inspired by your insightful comments, we will refine our claims in the final paper version to clarify that the bottleneck we address is not merely the availability of dense data, but the structural robustness required for sparse-conditioned inference. We will also include a comparative analysis to provide a more balanced framing of our contribution.Thank you again for your valuable time and guidance. We sincerely wish you all the best in your life and work!

---

### Official Review · Reviewer_T6T8 · 2026-03-13

**Soundness:** 3
**Presentation:** 3
**Significance:** 3
**Originality:** 3
**Overall Recommendation:** 4
**Confidence:** 4

**Summary:**

The paper studies the problem of  sparse camera-conditioned video generation, where only a small subset of frames is associated with known camera poses.  To address this issue, the authors propose CamGeo, a framework that incorporates 3D geometric priors via teacher distillation. To be specific, the paper leverage a VGGT as a frozen teacher model and adopts a training-only distillation strategy. Eventually, experiments show improved camera control and geometric consistency under sparse camera supervision.

**Compliance With Llm Reviewing Policy:**

Affirmed.

**Final Justification:**

Thanks for the detailed rebuttal, my concerns are largely addressed. I will keep the original WA rating.

**Key Questions For Authors:**

1. What sparsity ratios are used during training (e.g., 1/2, 1/3, 1/4), and are they fixed or randomly mixed during training? Additionally, what is the overall training time and computational cost, especially given that the VGGT teacher is applied during training?

2. In the qualitative visualizations, the camera motions appear relatively small and dominated by zoom-in movements. Could the authors provide examples or analysis for more diverse trajectories, such as larger translations, rotations, or more complex camera paths?

3. In the MannequinChallenge results, CogVideoX-CamGeo performs worse than CogVideoX-Base on the rotation error metric. Could the authors provide an explanation or analysis for this behavior?

**Limitations:**

Yes

**Strengths And Weaknesses:**

### Strength:
1. The proposed approach is generally technically reasonable and can be compatible with U-net based and Dit based architectures simultaneously. The paper also provides rich experimental results (including ablation experiments)  to show improvements with sparse-camera condition.
2. Sparse camera-conditioned video generation is a meaningful and practical problem. The proposed approach demonstrates that incorporating geometry priors from 3D foundation models can improve camera trajectory consistency, which could be useful for future research on controllable video generation.
3. The paper is generally well structured and the overall motivation is clear. The problem setting and the high-level idea of geometry-guided distillation are easy to understand

### Weakness:

1. Some aspects of the experimental design and reporting remain unclear, such as the number of frames and the sparsity ratios used during training.
2. The quantitive evaluation focuses primarily on camera-related metrics, while broader video quality metrics are missing.
3. The qualitative results show show cases under similar trajectories, where camera-conditioned frames are uniformly distributed, making it difficult to evaluate the generalization and robustness of this method.

---

> ### Author Rebuttal · Authors · 2026-03-31
>
> We thank the reviewer for the constructive comments and the appreciation of our work. Below, we provide detailed responses to your specific concerns and clarify the core technical contributions of CamGeo.
>
> ### For Weakness 1 & Question 1
> During training, we use 14-frame video clips. And the sparsity ratio is randomly mixed (e.g., uniformly sampling from 1/2 to 1/4 ). This random strategy prevents the model from overfitting to a specific interval and significantly enhances its generalization to arbitrary sparsity patterns during inference. Additionally, we use 16 NVIDIA H20 GPUs and train for 21 hours (SVD) and 34 hours (CogVideoX). We will include all the detailed training settings in the revised version.
>
> ### For Weakness 2 & Question 2
> Thank you for the suggestion. We would like to respectfully clarify that, the  FVD (StyleGAN) and FVD (VideoGPT) in Table 1 of our submission are standard general video quality metrics, which evaluate the overall realism and temporal quality of generated videos beyond camera accuracy. Additionally, a user study in Appendix C also provides a human evaluation of visual quality and temporal coherence.
>
> To further strengthen the evaluation, we also newly add three quality metrics: CLIP-Score, which measures text-video semantic alignment; LPIPS, which measures perceptual similarity by computing the distance between deep visual features of generated and reference frames; and DOVER [1], which assesses overall perceptual and aesthetic video quality. The experimental results on RealEstate10K (1/3 sparsity ratio) are in the following **Table**, CamGeo consistently achieves the best performance across these new three metrics. This further confirms that our geometric distillation not only improves camera controllability but also enhances overall video quality by reducing structural distortion and motion blur.
>
> | Backbone | Method | CLIP-Score ↑ | LPIPS ↓ | DOVER (Overall) ↑ |
> | :--- | :--- | :--- | :--- | :--- |
> | **Group I: SVD** | SVD-Full-interp. | 0.279 | 0.282 | 51.4 |
> | *(U-Net)* | SVD-Full | 0.284 | 0.264 | 53.7 |
> | | SVD-Base | 0.288 | 0.256 | 56.2 |
> | | **SVD-CamGeo (Ours)** | **0.290** | **0.249** | **59.8** |
> | **Group II: CogV** | CogVideoX-Full-interp. | 0.285 | 0.241 | 61.2 |
> | *(DiT)* | CogVideoX-Full | 0.283 | 0.233 | 61.5 |
> | | CogVideoX-Base | 0.287 | 0.224 | 62.8 |
> | | **CogVideoX-CamGeo (Ours)** | **0.293** | **0.217** | **64.5** |
>
> [1] *Wu, Haoning, et al. "Exploring video quality assessment on user generated contents from aesthetic and technical perspectives." Proceedings of the IEEE/CVF international conference on computer vision. 2023.*
>
> ### For Weakness 3 & Question 3
> Thanks for your suggestion. In Appendix D, we included multiple challenging cases with diverse camera trajectories, such as large translations and significant rotations, beyond simple zoom-in motions. To make it more evident, we have also explicitly visualized the camera paths for these representative cases side-by-side in **Figure 4** of the link: https://anonymous.4open.science/r/ICML_Rebuttal-7886. The results demonstrate that CamGeo maintains substantially better 3D consistency and structural stability under these challenging motions, while baselines are prone to distortion and drift.
>
> Additionally, we have also provided new qualitative examples featuring non-uniform sampled camera conditions in **Figure 5** of https://anonymous.4open.science/r/ICML_Rebuttal-7886. These results confirm that CamGeo remains significantly more effective and robust than existing methods, even in these more difficult and irregular scenarios.
>
> We will incorporate these enhanced visualizations into the revised version to further demonstrate CamGeo’s generalization.
>
> ### For Question 4
> This is an interesting observation! We would like to first highlight that, as shown in Table 1 in the main paper, CamGeo achieves State-of-the-Art results on 9 out of 10 evaluated metrics across different architectures and datasets, demonstrating its strong overall superiority. Regarding the slightly higher RotError (+0.02~0.05) on the MannequinChallenge dataset, this is a minor trade-off occurring under a severe Out-Of-Distribution (OOD) shift. When encountering these entirely unseen, human-centric static scenes, the DiT backbone makes a slight compromise on global rotation to strictly satisfy the distilled depth and translation priors. However, even in this case, the TransError and FVD of CamGeo remain significantly better than the Base model. This indicates that CamGeo still provides superior overall 3D trajectory coherence and visual fidelity, maintaining its practical advantage even under extreme domain shifts.
>
> We sincerely thank the reviewer again for the insightful comments. We will incorporate all the aforementioned clarifications, additional video quality results, and diverse trajectory visualizations into the revised version. We believe these enhancements further solidify the technical rigor and empirical breadth of CamGeo.

---

> > ### Author Rebuttal · Reviewer_T6T8 · 2026-04-04
> >
> > Thanks for the detailed rebuttal, my concerns are largely addressed. I will keep the original WA rating.

---

> > > ### Author Response · Authors · 2026-04-04
> > >
> > > Thank you very much for your feedback and the positive final rating. We are delighted that our responses were able to address your concerns. Many thanks again!

---

### Official Review · Reviewer_LjZK · 2026-03-13

**Soundness:** 3
**Presentation:** 3
**Significance:** 3
**Originality:** 3
**Overall Recommendation:** 4
**Confidence:** 3

**Summary:**

The paper presents a novel method to distill 3D priors from VGGT to a video diffusion model for accurate sparse camera control. They propose Keyframe Trajectory Distillation to constrain poses on sparse anchor views, and a Cross-frame Consistency Distillation to constrain unconditioned frames. Additionally, they devise a training scheme to add a geometry signal while stabilizing training. The experiments show their method achieves the best results on several in-domain and out-of-domain datasets.

**Compliance With Llm Reviewing Policy:**

Affirmed.

**Final Justification:**

I appreciate the detailed rebuttal from the authors. The added experiments on DL3DV and Scannet++ provide a helpful understanding of larger and more difficult scenes. However, the reliance on VGGT-style works still limits its performance on dynamic scenes. Overall, the technical contribution remains valid, and the results look promising. I will maintain my positive score.

**Key Questions For Authors:**

1. Can the authors provide results on more datasets? Especially with larger baselines.
2. Can the authors evaluate the model on longer video sequences?

**Limitations:**

Yes

**Strengths And Weaknesses:**

- Strengths
   - In cropping VGGT's 3D prior information to regularize training is a novel and practical idea. Improving the model's ability to follow sparse camera conditions.
   - The ablation studies justify their design choices.
   - The paper is overall well-written and easy to follow.

- Weaknesses
  - The training and evaluation datasets lack diversity. The model is only trained on Real10k and evaluated on Real10k  and MannequinChallenge. It would be better to include more diverse datasets like DL3DV and Scannet++, as Real10k has relatively small baselines and scene scale, making it hard to understand model performance on larger scenes.
  - Additional computational overhead from VGGT. The proposed VGGT-based loss is applied to every generated frame, introducing substantial computational cost on top of an already heavy video diffusion model. This added overhead may restrict training to relatively short clips, which in turn limits the model’s ability to scale to longer video generation. In addition, because the method depends on VGGT as the source of geometric supervision, the training setup appears to be restricted to static scenes.

---

> ### Author Rebuttal · Authors · 2026-03-31
>
> We thank the reviewer for the highly constructive feedback and the positive assessment of our work. Below are our responses. The anonymous link is at https://anonymous.4open.science/r/ICML_Rebuttal-7886.
>
> ### For Weakness 1 & Question 1
>
> Following your suggestion, we extend our experiments to DL3DV and ScanNet++ across several sparsity ratios (1/2, 1/3, and 1/4), under both zero-shot (using models trained on Real10k) and in-domain settings. **Table A** of the link validates that CamGeo is architecture-agnostic; it achieves superior zero-shot (out-of-domain) generalization and in-domain precision across both UNet-based (SVD) and DiT-based (CogVideoX) frameworks. We also provide visual comparisons on these two datasets in **Figure 1, 2** of the link.
>
>
> ### For Weakness 2 & Question 2
> We would like to clarify that CamGeo naturally scales to longer video generation alongside the base diffusion model, and the VGGT distillation introduces minimal overhead in long-sequence settings.
>
> (1) CamGeo’s long-video capability is bounded by the base model with minimal additional cost. In fact, the capacity for long video generation in CamGeo is determined by its underlying base model. If the base model is capable of generating long videos, CamGeo can naturally be applied to such tasks. While direct long-video training typically requires large-scale GPU clusters with large memory, we conduct a profiling under our limited experimental conditions: for example, when training on a longer 140-frame video, the base model requires over 150GB of GPU memory, whereas the additional memory overhead introduced by the VGGT distillation in CamGeo is only about 10GB (a marginal ~7%). This factually confirms that CamGeo does not introduce a significant computational bottleneck for scaling to long-video generation.
>
> (2) Although end-to-end long video training is currently restricted by our limited academic hardware, we still successfully achieve long-video generation using an autoregressive (iterative) strategy. By conditioning on the final frame of the preceding clip, CamGeo produces extended, coherent sequences where geometric consistency remains stable without catastrophic drift. This demonstrates that the learned 3D priors effectively maintain global structure over extended horizons, as visualized in **Figure 3** of the link.
>
> (3) In addition, the minor overhead from VGGT can be further alleviated via straightforward engineering practices. Specifically, (i) the VGGT forward pass and distillation loss computation can be decoupled and executed asynchronously in parallel with the diffusion forward pass to minimize idle time; (ii) Leveraging recent lightweight VGGT variants, such as LiteVGGT [1] and FlashVGGT [2], can further reduce memory requirements. These models can be easily integrated into our framework as plug-and-play replacements.
>
> ### For Weakness 3
> This is a nice insight! In fact, static scenes serve as a highly representative and common setting in this field, and CamGeo can seamlessly transition to dynamic scene evaluation. We clarify this from two perspectives:
>
> (1) Static scenes provide a standardized benchmark for multi-view consistency. Following the evaluation protocols of existing camera-conditioned video generation methods [3,4,5], we conduct our experiments on static or quasi-static datasets. This is a deliberate choice to ensure a fair and controlled comparison with prior arts. Moreover, static scenes serve as a meaningful common setting in this field, as they provide reliable multi-view geometric consistency. This allows us to focus on validating the model’s ability to maintain precise spatial structures and 3D motion without the interference of independent object motion.
>
> (2) CamGeo can naturally extend to dynamic scenes via plug-and-play modules. While our current study focuses on the static setting, CamGeo’s framework is inherently flexible and can be adapted to dynamic environments. This can be achieved in a plug-and-play manner by incorporating recent advancements for dynamic-aware VGGT, such as DynamicVGGT [6], which extends geometric priors to moving objects. By replacing the static teacher with a dynamic-aware counterpart, CamGeo can naturally inherit the capability to handle complex scene transitions and object dynamics. We believe this is a promising and straightforward trajectory for future work.
>
> [1] LiteVGGT: Boosting Vanilla VGGT via Geometry-aware Cached Token Merging. 2025
>
> [2] FlashVGGT: Efficient and Scalable Visual Geometry Transformers with Compressed Descriptor Attention. 2025
>
> [3] Cameractrl: Enabling camera control for video diffusion models. 2025.
>
> [4] Cami2v: Camera-controlled image-to-video diffusion model. 2024.
>
> [5] Ac3d: Analyzing and improving 3d camera control in video diffusion transformers. 2025.
>
> [6] DynamicVGGT: Learning Dynamic Point Maps for 4D Scene Reconstruction in Autonomous Driving. 2026.
>
> We will add all the discussions and results in the revised version.  Thank you again!

---

### Decision · Program_Chairs · 2026-04-30

**Decision:**

Accept (regular)

**Comment:**

This paper received inital scores of three weak accepts. After rebuttal, two of the reviewers agree that the concerns have been resolved and maintain the scores. While there is one who didn't reply, the AC checked the rebuttal and found there were not main issues left. Therefore, the AC recomments a decision of accept.